# Catholic Religious Practices Questionnaire (CRPQ): Construction and Analysis of Psychometric Properties

**Dariusz Krok [1,*]**, **Małgorzata Szcześniak [2]**, **Adam Falewicz [2]** and **Janusz Lekan [3]**

[1] Institute of Psychology, University of Opole, 45-040 Opole, Poland
[2] Institute of Psychology, University of Szczecin, 71-017 Szczecin, Poland
[3] Faculty of Theology, John Paul II Catholic University of Lublin, 20-950 Lublin, Poland
* Correspondence: dkrok@uni.opole.pl

**Abstract:** Members of the Catholic Church express their faith in a variety of manners, in general with a focus on liturgical and popular forms of piety. This article provided construction and initial validation for a brief questionnaire to measure Catholic religious practices. The authors used Sample 1 (*n* = 219) for exploratory factor analysis and Sample 2 (*n* = 181) for confirmatory factor analysis and to test the validity of a new scale. A model with two factors with five items each provided a good fit. The Catholic Religious Practices Questionnaire (CRPQ) consists of two subscales: official religiosity and folk practices. Both exhibit positive though varying correlations with the Centrality of Religiosity Scale (CRS) and Multidimensional Prayer Inventory (MPI). The new questionnaire has been confirmed as a reliable and valid measure that takes into account the distinctive features of the Catholic religious tradition.

**Keywords:** religious practices; Catholic Church; psychological assessment; psychology of religion





## 1. Introduction

The phenomenon of religiosity constitutes a complex and multifaceted reality that is investigated on different levels of scientific reflection: theological, philosophical, psychological, and sociological levels. This is due to the fact that religiosity comprises such forms as religious beliefs, personal and communal worship, religious experience, or emotions, which manifest themselves in different spheres of personal and social life (Głaz 2021a; Koenig et al. 2015). The aforementioned scientific disciplines are based on their own assumptions and use specific methods to describe and determine the phenomenon of religiosity. The recent polls conducted by the Gallup Poll Social Series (GPSS) found that around 81% of Americans believe in God, and 42% are convinced that God hears their prayers (The Gallup Poll Social Series 2022). In Poland, 81% of the population regard themselves as believers or firm believers, 46% attend a mass or a religious service at least once a week, and around 70% pray daily or at least once a week (Statistics Poland 2019). The development of new objective and constructive measures of religious practice is, therefore, integral to a psychological understanding of the role played by religion in contemporary life (Hill 2013). Despite the growing literature on religious practices, few reported psychological measures refer specifically to Catholic religious practices.

The aim of the present study is to develop a reliable and valid questionnaire that would assess Catholic religious practices conceptualized within the dogmatic and liturgical norms of the Catholic Church. Within this objective, we will assess the psychometric qualities of the questionnaire: the factor structure through exploratory factor analysis, the association between manifest variables and latent factors through confirmatory factor analysis (CFA), internal reliability, and convergent validity.

## 2. Theoretical Background

### 2.1. Specific Features of Catholic Religious Practices

Catholicism belongs to a common stream of Christianity based on dogmatic and doctrinal principles derived from the Holy Scriptures, the teachings of Christ, and the Tradition. The Roman Catholic Church has developed a highly advanced theology and an elaborate organizational structure since its beginning nearly two thousand years ago (Morrill 2021). The Catholic spiritual life revolves around the seven sacraments which are specific rituals through which believers receive God's grace: baptism, reconciliation, Eucharist, confirmation, marriage, holy orders, and the sacrament of the sick. They form the core of religious practices which play an important role in the everyday life of the Catholic Church.

Although the vast majority of world religions emphasize the need to practice faith, one of the characteristic features of Catholicism is the strong emphasis on the necessity of religious practices. The faithful are expected to attend mass on Sundays and holy days of obligation, go to Confession at least once a year, receive Holy Communion at least at Easter, and observe the prescribed days of fasting and abstinence (Catholic Church 2003). The traditional Roman Catholic view emphasizes that the Christian life should include the active expression of one's faith through religious practices, through which a person develops their faith, builds an inner spiritual life, and becomes an increasingly conscious and responsible Christian. Faith is only alive and strong when it is actively professed and practiced. Thus, within the Catholic tradition, religious practices are seen as visible signs and personal externalizations of internal religious beliefs and feelings (Cush 2020). Through rituals and forms of worship, individuals express their faith as well as deepen it.

Although the participation of Catholics in worship is circumscribed by the relevant religious and ecclesiastical norms, including the sanction of grave sin, these norms can be accepted by them to varying degrees and assessed from different points of view. In fact, religious faith and practices are strongly interconnected with cultural and social factors, which often determine the frequency of Mass attendance and prayers or the observance of fasting (Allison 2014). Religious practices are motivated by various factors, not always of a religious nature, as one may engage in religious practices due to the demands of the dominant culture or the pressures of the family environment (Ganiel 2019; Giordan et al. 2018); in that case, religiosity takes on merely a socio-cultural character. This approach is visible in the new SpREUK-P questionnaire which measures Christian religious practices, especially those which are specific to Catholic rituals and practices (Büssing et al. 2017). It was also validated in Polish on a group of patients with chronic diseases (Büssing et al. 2014). The questionnaire demonstrated the need to distinguish Christian practices, rituals, and behaviors, and examine them in the context of one's own denomination.

From a sociological point of view, religious practices are regarded as the religious behaviors of members of a religious group (Day 2020). These may take the form of communal or private practices. The most important religious practices in the Catholic Church include Holy Mass, Devotions to Jesus Christ (e.g., Eucharistic adoration, the Divine Mercy Devotions), novenas, various litanies (e.g., to Our Lady, to saints), the Stations of the Cross, prayers (e.g., the Angelus, the Rosary), and Devotions to saints (e.g., prayers of intercession). The research demonstrated that it is rarely that cultic practices are driven by purely religious motives. In most cases, they are inspired by "mixed motives" which are the consequence of the interaction of several motives, e.g., psychological, social, or cultural ones (Park 2021). Their multidimensional character does not devalue them because religious factors are always embedded in a specific culture and one's personality. The focus of this paper will be on those religious practices that belong to the ritual dimension of the Catholic Church and express the spiritual character of the faith.

### 2.2. Psychological Measurement of Religious Practices

The study of religious practices is particularly important in the psychological explanation of the consequences of professed faith. First, reliable opinions about religiosity

cannot be formulated without an adequate diagnosis of religious practices. The research clearly demonstrated that the ways in which individuals practice their religion influence their overall religiosity, including their thinking and feelings expressed towards the sacred (Hobson et al. 2018; Van Tongeren et al. 2021) and personal relationship with God (i.e., religious experiences) (Głaz 2021b). Second, religious practices are related to a number of psychosocial factors, e.g., personality, coping with stress, or social norms and values (Krok et al. 2021; Meuleman and Billiet 2018; Stronge et al. 2021). The psychological approach to religious practices involves the study of the structures, formative processes, and functions that religious elements perform in the cognitive, emotional, and social dimensions of one's functioning. The religious practices that an individual perceives and internalizes play an important role in their life, by modifying their thinking, the emotions they experience, and their behavior both toward themselves and other people.

Religious practices have been measured from different methodological perspectives. Analyzing the available measures of religiosity (Hill 2013), the following categories assessing religious practices can be listed: (1) scales that assess religious or spiritual commitment, (2) scales that assess religious social participation or religious/spiritual support, (3) scales that assess private religious or spiritual practices, and (4) scales that assess religious or spiritual experiences. Due to space constraints in this article, only a selection of the most important examples of scales measuring religious practices will be presented. The Religious Commitment Inventory-10 (Worthington et al. 2003) can serve as an example of the first of the aforementioned categories. The inventory consists of 10 items that reflect religious commitment which is understood as the degree to which individuals adhere to their religious values, beliefs, and practices and apply them on a daily basis. The underlying assumption is that highly religious people tend to perceive the world through religious schemas and consequently will attempt to live in accordance with their religion. Therefore, religious practices will directly or indirectly reflect one's religious convictions to a certain degree. The research showed that religious commitment was positively related to self-efficacy, marital adjustment, dispositional interpersonal forgiveness, and lower rumination, and negatively related to academic dishonesty (Lopez et al. 2011; Onu et al. 2021; VanOyen et al. 2008).

Religious practices are also manifested in the degree to which people engage in religious activities and behavior. The Religious Involvement Inventory (Hilty and Morgan 1985) measures one's involvement in religious activities or practices. It comprises 82 items that form 7 different dimensions of religiosity: personal faith, intolerance of ambiguity, orthodoxy, social conscience, knowledge and religious history, life purpose, and church involvement. Studies using this inventory demonstrated that for Catholics, personal faith, orthodoxy, and church involvement were positively associated with life satisfaction and positive affect, whereas for Pentecostals, these three subscales were positively associated with life satisfaction and positive affect and, in addition, negatively associated with negative affect (Chamberlain and Zika 1992). In addition, religious involvement was related to lower levels of depression, hopelessness, and suicidal ideation among high school students (Gray 2004).

A widely used scale that assesses religious or spiritual private practices is the Centrality of Religiosity Scale (Huber 2006) which measures the importance and content of the religious construct system in religious practices of theistic religions. The scale consists of 15 items that represent 5 dimensions: (a) cognitive interest—it denotes the intensity and strength of intellectual interest in religious issues, (b) ideology—it describes ideas related to God's existence, religious beliefs, and doctrines, (c) prayer—it assesses the frequency of religious rituals and behavior, and (d) religious experience—it scrutinizes individuals' spiritual relationships with God, and worship, and it reflects the level of attendance at church services. The last three dimensions directly represent religious practices, either private (prayer, religious experience) or public (worship). The research conducted predominantly among Catholics demonstrated that the centrality of religiosity was positively related to empathy and exposure to credible religious acts during childhood (Łowicki and Zajenkowski 2019), authoritarianism (Krok 2011), sacred values, and, to some extent, the

value of truth (Krok and Cholewa 2021). The dimensions of centrality also had associations with behavioral procrastination, and these relationships were mediated by the locus of control (Zarzycka et al. 2021). These results confirm that religious practices, being deeply embedded in a religious realm, play a vital role in people's behavior.

The Scale of Personal Religiosity (Jaworski 1989), developed to measure Catholic aspects of religiosity, represents the last of the above categories: scales that assess religious or spiritual experiences. It aims to measure religious attitudes and differentiate between personal and impersonal religiousness. Personal religiosity describes deeply rooted and internalized beliefs in God, which are expressed through religious practices, spiritual development, membership in a church community, and adherence to religious rules in daily life. Impersonal religiousness is the opposite of the previous attitude as it reduces faith only to the observance of traditions, rituals, and superficial forms of piety without any deeper awareness of a personal relationship with God. The scale contains 30 items that represent 4 dimensions: religious faith, morality, religious practices, and the religious self. Examining a sample of Catholics, Noworol and Głaz (2021) showed that religious practices were positively associated with two personality traits: openness and conscientiousness. Religious practices were also positively related to the level of marital satisfaction among Catholic spouses, specifically to intimacy (i.e., a close relationship between spouses), resemblance (expressing harmony between spouses in relation to important goals in family life), and self-realization (perceiving marriage as a relationship that enables each partner to realize themselves) (Gosztyła and Gelleta 2015). As can be seen, religious practices are, therefore, linked to both personal and relational factors.

Taken together, these findings provide empirical evidence pointing out that religious practices firstly are a distinct dimension of religiosity. Secondly, they play an important role in the overall religiosity of individuals, and thirdly, they are interconnected with psychosocial functioning. It thus becomes fully justifiable to attempt to construct a scale that measures religious practices conducted within Catholicism and which could, in the future, be used to investigate the relationship of religious behavior with psychological and social factors.

### 2.3. Overview of the Present Research

The study aimed at firstly developing a reliable, brief, and valid questionnaire that would assess Catholic religious practices as they are understood within the dogmatic and liturgical norms of the Catholic Church; secondly, establishing the factor structure through exploratory factor analysis (EFA, Study 1); thirdly, assessing the association between manifest variables and three latent factors through confirmatory factor analysis (CFA); and fourthly, estimating the internal reliability of the CPRQ and evaluating its convergent validity. In light of the previous research (Hilty and Morgan 1985; Huber 2006; Van Tongeren et al. 2021), the study presented in this paper assumed that the questionnaire would have a two-dimensional structure: official religiosity (it represents the extent to which believers shows compliance with the demands of the Catholic Church in terms of practice and observance of the official teaching and the acceptance level of religious symbols and teachings in the public space) and folk practices (the subjective importance of customary but informal forms of prayer called popular Catholic piety). The authors also hypothesized that subscales would demonstrate good reliability and correlate with other religiosity indicators.

### 3. Study 1

#### 3.1. Development of Original Item Pool

The aim of Study 1 was to generate a pool of statements that were related to a preliminary conceptual definition of religious practices in the context of liturgy and folk piety. The identification process was based on a review of the literature (deductive method) and responses from individuals (inductive method) as such an approach is regarded as best practice in item development (Boateng et al. 2018). The *Directory on Popular Piety and Liturgy*

was one of the main sources of inspiration for creating items. In addition, we included people's views on their practice of faith. Due to the specificity of the planned questionnaire, we did not use other statements from the existing scales. In this sense, all items were originally created by the research team. The main criterion adopted in the creation of items was a reference to the manifestations of official religiosity (e.g., liturgy) and folk piety (e.g., cultural elements), understood as complementary forms of expressing faith.

Following the guidelines that the initial item pool should contain at least twice as many items as the intended questionnaire (Morey 2013), the authors of this study created 40 stimulus items. They also tried to keep the content of the items simple and non-ambivalent (Boateng et al. 2018), and included reverse-scored items to reduce some potential acquiescence bias (Field and Miles 2010). Four experts in the field of theology and psychology examined the congruency of items. They evaluated all items considering the compliance of the content of each statement with the criterion referring to official religiosity and popular piety. They also paid a special attention to the character of Catholic traditions. Based on their evaluations, the pool was reduced to 15 items (Table 1).

**Table 1.** Item pool of 15 after expert evaluation (Sample 1).

| Items | Content |
|-------|---------|
| CRPQ1 | I avoid going to Mass where priests threaten me with hell and the punishment for sins (Reversed). |
| CRPQ2 | The forms of popular piety express faith in God in a simple way. |
| CRPQ4 | Popular piety is a legitimate expression of the Christian faith. |
| CRPQ7 | All religions carry the same values as secular worldviews (Reversed). |
| CRPQ8 | Religious images in public areas are part of popular tradition and should be kept there. |
| CRPQ10 | I hardly ever pay attention to official Church teachings (Reversed). |
| CRPQ14 | I try to attend Mass every Sunday. |
| CRPQ18 | It does not really matter whether I believe in the Christian or Muslim God, in many gods or none (Reversed). |
| CRPQ19 | Public life must be free from the influence of the Church (Reversed). |
| CRPQ22 | Participation in Holy Mass is the most important element in the practice of my faith. |
| CRPQ26 | The practices of popular piety contribute to growth in faith. |
| CRPQ29 | The moral principles proposed by the Church are not adapted to the current reality (Reversed). |
| CRPQ35 | Simple popular religious practices express the depth of faith. |
| CRPQ38 | Religious symbols such as crucifixes or images of the Virgin Mary should not be seen in public (municipal offices, public services, public schools, and hospitals) (Reversed). |
| CRPQ39 | The religiosity of the people is testimony to the faith of people with simple hearts. |

All items were assessed on a 7-point Likert scale with 1 = strongly disagree; 2 = disagree; 3 = partially disagree; 4 = neither agree nor disagree; 5 = partially agree; 6 = agree; and 7 = strongly agree. In terms of content, the statements reflected the two-dimensional structure of the CRPQ, referring to folk practices and official religiosity.

### 3.2. Participants

Sample 1 comprised 219 participants between 18 and 87 ($M$ = 26.92; $SD$ = 11.39) and included 146 women (67%) and 73 men (33%). In terms of place of residence, 25% of respondents indicated the countryside, 17%—a city up with to 25,000 inhabitants, 13%—a city between 25,000 and 100,000 inhabitants, and 25%—a city above 100,000 inhabitants. When asked to what extent they believe that God exists, the average obtained was $M$ = 7.10 ($SD$ = 3.34), on a scale from 0 to 10. The connection with the Catholic Church was at the level of $M$ = 6.00 ($SD$ = 3.72). When it regarded participation in the Holy Mass, most people (43%) indicated the Sunday Mass, followed by those who go to Mass only on major liturgical feasts

(22%), every day (7%), and once every few weeks (6%). Exactly 22% of the respondents declared that they never attended Holy Mass. When asked whether they belong to any group related to the Catholic Church, the respondents who answered positively, indicated the following communities: University Chaplaincy Centre, Light-Life Movement, Liturgical Service of the Altar, Living Rosary, Priesthood, Religious Congregation, Evangelizing Community, Catholic School, Catholic Youth Association, Parish Pastoral Care, Diocesan Diaconia, Caritas Polska, Vincentian Marian Youth, Bible Circle, and Church Choir.

Participants were recruited through purposive sampling, using different internet-based networks and email correspondence that was addressed to individuals who declared their affiliation with the Catholic Church.

### 3.3. Procedure and Data Analysis

A preliminary procedure was performed to implement exploratory factor analysis (EFA). All CRPQ items were assessed for normality with skewness and kurtosis statistics. Acceptable limits of $\pm 2$ were assumed (Field 2009).

As there is no single subject-to-item ratio for developing a questionnaire (Boateng et al. 2018), a ratio of 5:1 in EFA (Gorsuch 1983; Costello and Osborne 2005) was assumed. It is a commonly used ratio of observations to variables (Osborne and Costello 2004) which does not influence factor stability (Arrindell and van der Ende 1985). The sample of 400 observations was randomly divided in two, with $n = 219$ for Study 1 (EFA) and $n = 181$ for Study 2 (CFA). There were no missing values.

Next, the EFA was carried out with a maximum likelihood (ML) estimation, eigenvalues > 1, and promax rotation as the factors were expected to be correlated. The authors used the Kaiser–Meyer–Olkin measure of sampling adequacy with a cut-off of 0.80 considered meritorious (Hutcheson and Sofroniou 1999) and Bartlett's test of sphericity value of $p < 0.05$ (Hair et al. 2019a, 2019b). Additionally, a scree plot was examined as a good graphical measure for determining the number of underlying factors. The total variance was assumed at the level of 56.6%, following Peterson's results based on the meta-analysis of behavioral data (Peterson 2000). The authors considered at least 5% (Hair et al. 2019a, 2019b) for a second factor and items with loadings above 0.63 (Tabachnick and Fidell 2013) to ensure a more reliable and robust questionnaire (Szcześniak et al. 2022). The reliability of expected factors was measured, assuming a value of $\alpha > 0.70$ as an acceptable indicator of the internal consistency of the CRPQ.

As in the final selection of the proper items, it is preferable to take into account various indicators, the next computed statistic was the corrected item–total correlation, which shows consistency between an item and other items in a factor (Zijlmans et al. 2019). The range of item–remainder correlations (Bandalos 2018) was considered acceptable between 0.30 and 0.70.

The current research project was approved by the Research Ethics Committee of the John Paul II Catholic University of Lublin (Institute of Theological Sciences), number KEBN_31/2022. All statistical analyses were computed with the use of IBM SPSS statistics package version 20 and IBM SPSS AMOS 21.

### 3.4. Results

The descriptive statistics showed an approximately normal distribution as all items tested ranged between $-2$ and $+2$ for the values of skewness and kurtosis (Table 2).

The indices of the appropriateness of the data for factor extraction confirmed the adequacy of the sample to conduct such analysis. The Kaiser–Meyer–Olkin statistics showed a value of 0.911 indicating no problem with the sample size. The Bartlett Test of Sphericity of all fifteen items displayed acceptable correlations to continue the factorial analysis ($\chi 2 = 1817.448$, df = 105, $p < 0.001$). The EFA with the maximum likelihood (ML) estimation (unforced promax rotation) showed two components with eigenvalues greater than 1.0 which accounted for 58.5% of the variance. The graphical presentation (a scree plot) also suggests a two-factor solution for the CRPQ (Figure 1).

**Table 2.** Descriptive statistics for items of the CRPQ (Sample 1).

| Items | M | SD | Min | Max | Skewness | Kurtosis |
|---|---|---|---|---|---|---|
| CRPQ1 | 3.91 | 2.43 | 1 | 7 | 0.07 | −1.64 |
| CRPQ2 | 4.57 | 1.70 | 1 | 7 | −0.47 | −0.56 |
| CRPQ4 | 3.35 | 1.66 | 1 | 7 | 0.31 | −0.63 |
| CRPQ7 | 2.80 | 1.75 | 1 | 7 | 0.74 | −0.42 |
| CRPQ8 | 4.58 | 1.94 | 1 | 7 | −0.37 | −1.01 |
| CRPQ10 | 3.18 | 2.19 | 1 | 7 | 0.55 | −1.18 |
| CRPQ14 | 4.15 | 2.70 | 1 | 7 | −0.06 | −1.84 |
| CRPQ18 | 3.35 | 2.35 | 1 | 7 | 0.46 | −1.38 |
| CRPQ19 | 4.10 | 2.29 | 1 | 7 | −0.05 | −1.53 |
| CRPQ22 | 3.91 | 2.50 | 1 | 7 | 0.07 | −1.70 |
| CRPQ26 | 3.85 | 1.54 | 1 | 7 | −0.11 | −0.35 |
| CRPQ29 | 3.91 | 2.50 | 1 | 7 | 0.07 | −1.70 |
| CRPQ35 | 4.16 | 1.73 | 1 | 7 | −0.23 | −0.68 |
| CRPQ38 | 3.32 | 2.09 | 1 | 7 | 0.52 | −1.00 |
| CRPQ39 | 4.44 | 1.67 | 1 | 7 | −0.35 | −0.40 |

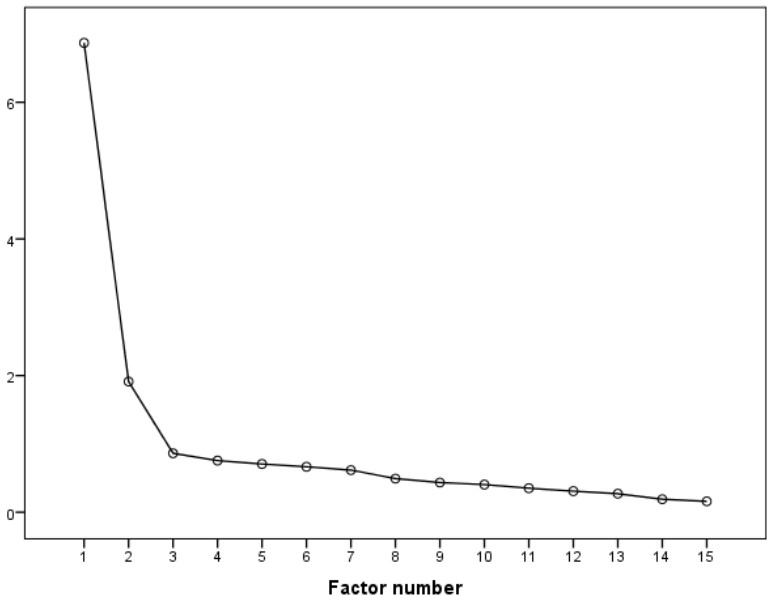

**Figure 1.** Scree plot.

All items showed loadings greater than 0.6, except for item CRPQ8, and a very good internal consistency (Factor 1—official religiosity $\alpha$ = 0.917 and Factor 2—folk practices $\alpha$ = 0.816). Based on this criterion, item CRPQ8 was excluded from further analysis (Table 3). The percentage of variance explained for official religiosity was 45.797 and for folk practices was 12.752.

The corrected item–total correlations were between 0.429 and 0.746 for items of official religiosity and between 0.530 and 0.617 for items of folk practices. Therefore, to avoid the risk of redundancy, the item with the highest value of 0.746 (CRPQ14) was removed. Thus, the following items were accepted for the CFA analysis in Study 2: official religiosity—CRPQ1, CRPQ7, CRPQ22, CRPQ29, CRPQ38 and Folk practices—CRPQ2, CRPQ4, CRPQ26, CRPQ35, CRPQ39. The Cronbach alpha for official religiosity was $\alpha$ = 0.790, and for folk practices, it was $\alpha$ = 0.787.

**Table 3.** Promax rotation for items of the CRPQ (Sample 1).

| Items | Factor 1 | Factor 2 |
|---|---|---|
| CRPQ1 | **0.704** | 0.051 |
| CRPQ2 | 0.032 | **0.722** |
| CRPQ4 | −0.014 | **0.689** |
| CRPQ7 | **0.695** | 0.238 |
| CRPQ8 | −0.293 | **0.545** |
| CRPQ10 | **0.819** | −0.051 |
| CRPQ14 | **0.663** | 0.281 |
| CRPQ18 | **0.882** | 0.018 |
| CRPQ19 | **0.953** | 0.159 |
| CRPQ22 | **0.694** | 0.211 |
| CRPQ26 | 0.257 | **0.817** |
| CRPQ29 | **0.768** | 0.001 |
| CRPQ35 | 0.039 | **0.759** |
| CRPQ38 | **0.695** | −0.044 |
| CRPQ39 | −0.040 | **0.760** |

Legend: significant results for each factor are in bold.

## 4. Study 2

### 4.1. Participants

Sample 2 consisted of 181 adults between 18 and 82 ($M$ = 28.71; $SD$ = 11.87) and included 116 women (64%) and 65 men (36%). As Sample 2 is part of the entire pool of respondents, divided randomly for the purposes of EFA and CFA, the variables that were investigated are the same. With regards to the place of residence, 24% of respondents indicated the countryside, 15%—a city with up to 25,000 inhabitants, 12%—a city between 25,000 and 100,000 inhabitants, and 49%—a city above 100,000 inhabitants. When asked about belief in the existence of God, it was $M$ = 7.05 ($SD$ = 3.46). As for the relationship with the Catholic Church, it was at a level where $M$ = 5.55 ($SD$ = 3.78). When asked about participation in the Holy Mass, most people (38%) indicated the Sunday Mass, followed by those who go to Mass only on major liturgical feasts (22%), every day (8%), and once every few weeks (7%). Exactly 25% of the participants indicated that they never took part in Holy Mass. When asked if they belonged to any group associated with the Catholic Church, the respondents indicated: University Chaplaincy Center, Light-Life Movement, Liturgical Service of the Altar, Living Rosary, Priesthood, Religious Congregation, Evangelizing Community, Catholic School, Catholic Youth Association, Parish Pastoral Care, Diocesan Diaconia, Caritas Polska, Vincentian Marian Youth, Bible Circle, Foundation of Small Feet, Neocatechumenate, Home Church, Seminary, Religious Community, School of New Evangelization, Society of Saint Pius X, and Church Choir.

### 4.2. Procedure and Data Analysis

As the structural equation model has the assumption of normal distribution, before the application of CFA, the skewness and kurtosis of all ten items were tested following the guidelines of Tabachnick and Fidell (2013). Values ± 2 were assumed as a good approximation to normality.

In the next step, model parameters were examined in the CFA. Considering the thresholds of factor loadings reported by Harrington (2009, p. 23), the values "above 0.71 are excellent, 0.63 very good, 0.55 good, 0.45 fair, and 0.32 poor". The goodness-of-fit was evaluated using several widely known and common fit indices with values of an adequate model fit: adjusted to degrees of freedom (CMIN/DF, the goodness-of-fit index (GFI $\geq$ 0.90)); the Tucker-Lewis index (TLI $\geq$ 0.90); the comparative fit index (CFI $\geq$ 0.90), the standardized mean square residual (SRMS $\leq$ 0.08), the root mean square error of approximation (RMSEA $\leq$ 0.08; LO $\leq$ 0.08; HI $\leq$ 0.08) (Szcześniak et al. 2022).

Finally, the convergent validity was investigated to demonstrate whether and how the CRPQ corresponds to other measures of religiosity. To realize this goal, the Centrality

of Religiosity Scale (Huber and Huber 2012) and Multidimensional Prayer Inventory (Laird et al. 2004) were selected. The rationale for the use of both of the above-mentioned questionnaires was that they have a multivariate structure and different dimensions. Such a solution revealed the nuances of the newly created CRPQ.

*4.3. Measures*

The Centrality of Religiosity Scale (CRS) is a measure of the position of religious constructs, showing how central, important, or salient the religious meanings are within personality. The questionnaire consists of 15 items grouped into 5 subscales (Huber 2006; Huber and Huber 2012): intellect (measuring importance and frequency of cognitive analysis of religious themes), ideology (measuring subjective belief about the real existence of the transcendent reality and the level of openness to various forms of transcendence), private practice (measuring the frequency of actual attempts to contact with transcendence and its subjective significance), religious experience (measuring how often transcendence becomes an element of one's experience, and the person has a sense of God's presence or workings), and public practice (measuring the frequency and subjective importance of participation in religious services). The total score is the sum of the results on the subscales and provides an overall measure of the centrality of religious meanings in personality. The response options range from 1 (not at all/never) to 5 (to a great extent/very often). High scores on each subscale signify the high efficiency of the impact of religiousness on an individual's experiences and behavior. The scale presents satisfactory psychometric properties (Zarzycka 2011). The reliability of the CRS in the present study was $\alpha = 0.916$ for intellect, $\alpha = 0.902$ for ideology, $\alpha = 0.870$ for private practice, $\alpha = 0.910$ for religious experience, $\alpha = 0.882$ for public practice, and $\alpha = 0.957$ for the overall score.

The Multidimensional Prayer Inventory (MPI, Laird et al. 2004), in the Polish adaptation by Zarzycka et al. (2022) is a 15-item multidimensional self-report tool measuring types of prayer. It measures five traditional forms of prayer: adoration (e.g., I praised God), confession (e.g., I acknowledged faults and misbehavior), thanksgiving (e.g., I expressed my appreciation for my circumstances), supplication (e.g., I made various requests of God), and reception (e.g., I opened myself up to God for insight into my problems). The participants assess the extent to which they engage in particular types of prayer on a 7-point Likert scale (from 1 (never) to 7 (all of the time)). The reliability of the MPI in the current study was $\alpha = 0.870$ for adoration, $\alpha = 0.938$ for confession, $\alpha = 0.894$ for thanksgiving, $\alpha = 0.902$ for supplication, and $\alpha = 0.903$ for reception.

*4.4. Results*

Similarly to the findings in Sample 1, the descriptive statistics displayed a close-to-normal distribution as all items scored between −2 and +2 for values of skewness and kurtosis (Table 4).

**Table 4.** Descriptive statistics for items of the CRPQ (Sample 2).

| Items | M | SD | Min | Max | Skewness | Kurtosis |
|---|---|---|---|---|---|---|
| CRPQ1 | 4.41 | 2.45 | 1 | 7 | −0.26 | −1.57 |
| CRPQ2 | 4.44 | 1.93 | 1 | 7 | −0.47 | −0.81 |
| CRPQ4 | 3.40 | 1.93 | 1 | 7 | 0.28 | −0.98 |
| CRPQ7 | 5.22 | 1.86 | 1 | 7 | −0.78 | −0.50 |
| CRPQ22 | 3.44 | 2.36 | 1 | 7 | 0.29 | −1.55 |
| CRPQ26 | 3.46 | 1.62 | 1 | 7 | 0.07 | −0.56 |
| CRPQ29 | 4.33 | 2.28 | 1 | 7 | −0.23 | −1.42 |
| CRPQ35 | 3.97 | 1.91 | 1 | 7 | −0.15 | −0.94 |
| CRPQ38 | 4.64 | 2.33 | 1 | 7 | −0.46 | −1.33 |
| CRPQ39 | 4.33 | 1.76 | 1 | 7 | −0.36 | −0.61 |

The structure of the CRPQ was confirmed through the CFA. The loadings (Figure 2) were between 0.56 (good) and 0.87 (excellent) for official religiosity and between 0.62 (very good) and 0.72 (excellent) for folk practices. The goodness-of-fit showed that the two-factorial model consisting of official religiosity and folk practices adequately represented the data: CMIN/DF = 2.161; GFI = 0.93; TLI = 0.92; CFI = 0.94, SRMS = 0.05; RMSEA = 0.08; LO = 0.05; and HI = 0.10. Based on the results obtained, the model was accepted in its present form. The internal reliability for official religiosity was $\alpha = 0.838$, and for folk practices, it was $\alpha = 0.802$.

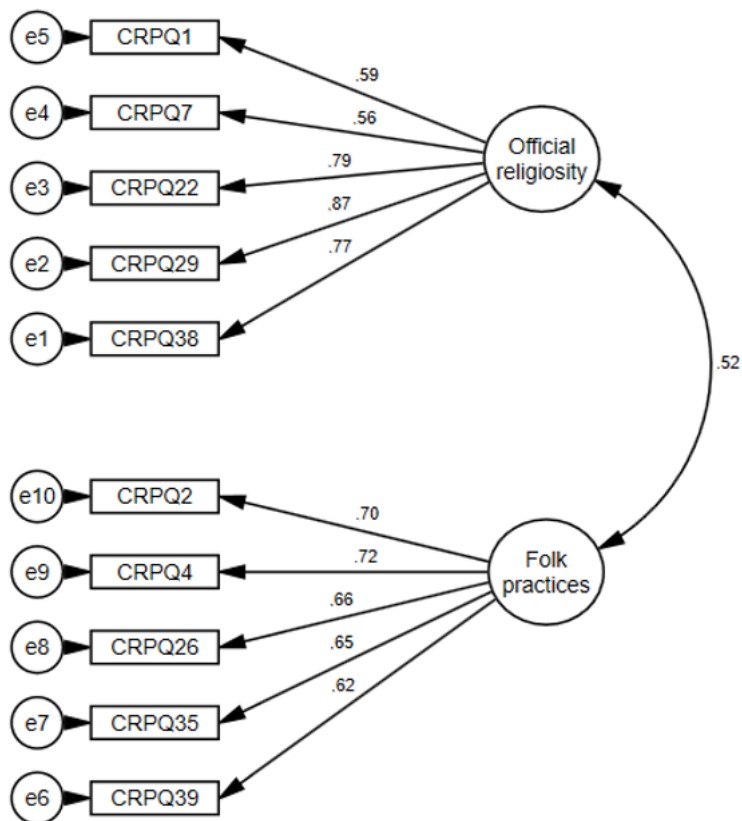

**Figure 2.** Measurement model of CRPQ.

Considering that the phenomenon of folk practices constitutes a multifaceted concept, we also tried to explore an alternative three-factor model. Although the loadings of the third factor that we called popular faith (CRPQ35 and CRPQ39) were satisfactory (between 0.66 and 0.78), the loadings of the two other factors decreased (two loadings were under 0.55). Moreover, the covariance between folk practices and popular faith exceeded 0.86, thus suggesting discriminant validity issues. The goodness-of-fit showed that the three-factorial model consisting of official religiosity, folk practices, and popular faith represented the data less adequately than the two-factorial model: CMIN/DF = 3.081; GFI = 0.87; TLI = 0.88; CFI = 0.91, SRMS = 0.07; RMSEA = 0.11; LO = 0.09; and HI = 0.13. Based on the results obtained, the two-factor model was accepted in its initial form.

Pearson's correlation values (Table 5) showed that both official religiosity and folk practices were positively and significantly associated with all the dimensions/overall score of the CRS and all dimensions of the MPI. Moreover, official religiosity correlated positively with folk practices at the level of $r = 0.43$ ***.

**Table 5.** Correlation values between official religiosity/folk practices, the CRS, and the MPI (Sample 2).

| Factors | Official Religiosity | Folk Practices | Corr. Value Difference |
|---|---|---|---|
| Intellect | 0.66 *** | 0.46 *** | 3.294 *** |
| Ideology | 0.63 *** | 0.49 *** | 2.284 * |
| Private practice | 0.71 *** | 0.46 *** | 4.308 *** |
| Religious experience | 0.57 *** | 0.44 *** | 1.994 * |
| Public practice | 0.79 *** | 0.54 *** | 4.981 *** |
| Overall score | 0.77 *** | 0.54 *** | 4.452 *** |
| Adoration | 0.57 *** | 0.33 *** | 3.716 *** |
| Confession | 0.53 *** | 0.27 *** | 4.036 *** |
| Thanksgiving | 0.40 *** | 0.26 *** | n.s. |
| Supplication | 0.33 *** | 0.29 *** | n.s. |
| Reception | 0.49 *** | 0.34 *** | 2.138 * |

Note. *** $p \leq 0.001$; * $p \leq 0.05$; n.s.—not significant.

Additionally, a two-tailed Fisher's z-test was conducted (Table 5) to examine the equality of two correlation coefficients obtained from the same sample. It allows a comparison of the values of dependencies between both CRPQ variables, i.e., the CRS and MPI variables (Lee and Preacher 2013; Steiger 1980).

## 5. Discussion

The study aimed to develop a Catholic Religious Practices Questionnaire, determine its internal structure and psychometric properties, and analyze its correlations with established measures of religiosity. To the authors' knowledge, the present paper is the first study to present a tool designed to measure Catholic religious practices, especially in relation to both official and folk dimensions.

The psychometric analyses in Study 1, consistent with the hypothesis formulated by the authors, suggested a two-factor structure for the CRPQ and very good internal consistency for both subscales. The first subscale, official religiosity (OR), refers to the extent to which a person shows compliance with the demands of the Catholic Church, referring to practice and observance of the official teaching. This dimension also assesses the level of consent to the presence of religious symbols and teachings in the public space. Official religiosity is not only a matter of the personal importance of attending religious practices but also obedience to the teachings of the Church. Therefore, it is necessary to understand that for a Catholic person, Mass is also an opportunity to profess faith and show that they identify with the group but also express an inner acceptance of the official teaching. That is why this factor combines both behavioral (participation) and cognitive (obedience to the Magisterium of the Church; cf. Sullivan 2002) aspects of Catholic practice. The specificity of this connection is present in one of the fundamental rules of Catholic Church theology—lex orandi lex credendi (De Clerck 1994). The principle ("the law of prayer [is] the law of belief") can have at least three meanings: (1) the way in which the Church prays expresses its faith; (2) liturgy is a form of expression of faith and, therefore, this form cannot be separated from the substance of faith; (3) the way of praying determines the way of believing (Ferdek 2012). The practice of officially defined forms of prayer is, therefore, inextricably linked to the content of religious beliefs. Four items in this scale are reverse-coded in order to avoid response style bias (Józsa and Morgan 2017; Suárez-Álvarez et al. 2018). High scores on the OR subscale reflect intensive cognitive and behavioral appreciation of the liturgy and official teaching of the Catholic Church. Low scores are connected to poorer attendance on Sunday Mass, lower observance of the teaching of the Church, and affirmation of the laicization processes in public sphere.

The second subscale, folk practices (FP), refers to the subjective importance of customary but informal forms of prayer known as popular piety, which are widespread among Catholics. This type of devotion is culture-dependent and can vary from country to country or even region to region. The dimension of folk practices refers to the belief in the importance of spontaneous forms of popular devotion in the development of faith and its

genuineness (Roszak and Tykarski 2020). Documents of the Catholic Church define the term "popular piety" as "diverse cultic expressions of a private or community nature which, in the context of the Christian faith, are inspired predominantly not by the Sacred Liturgy but by forms deriving from a particular nation or people or from their culture" (DPPL 2002, p. 9). Popular piety is, therefore, understood as "the form of Christianity in which devotional practices such as praying the rosary, going on pilgrimages, and venerating the Virgin Mary and the saints, occupy a central position" (Mong 2019, p. 2). It predominantly focuses on religious activities expressed by believers on a daily basis, which forms the core of their personal faith. High scores on the FP subscale show elevated appreciation of practice of the Catholic devotions and belief in the positive impact it has on the development of faith of the individual. Low scores mean rejection of spontaneous forms of piety and demeaning its role in the profile of prayer of a Catholic person.

The analyses in Study 2 have proven the factorial and convergent validity of both subscales. The obtained levels of internal reliability ($\alpha$ over 0.8) make this scale suitable for individual measurement purposes (Taber 2018). Although we also tested the three-factor model, the empirical evidence showed that the two-factor model represented a better solution. To verify nomological validity, a series of correlation analyses were conducted between the CRPQ and other measures of religiosity. The correlation matrix shows significant positive associations between both factors of CRPQ, that is CRS and MPI, which supports the initial hypotheses of this study.

The correlation level between OR and FP is moderate. It shows that, to a certain degree, they share the common scope of religious practices of Catholics but remain separate factors. It is even more visible when convergent validity is evaluated. Although both factors of CRPQ correlate positively with all dimensions and the CRS total score and MPI subscales, for the most part, the correlation coefficients differ significantly in magnitude in favor of official religiosity. In all cases, the OR subscale exhibits higher, from moderate (religious experience) to strong (public practice) correlation with all the dimensions/the overall score of the CRS. It suggests that the perception of religiosity understood as an official obligation is more closely related to the way a person sees religiosity as a central life issue. The strongest binding (0.79) connects OR with public practice (from CRS). It appears entirely intuitive that a person that perceives participation in religious services subjectively as a priority is also someone that observes participation in Mass on Sunday and so-called 'holy days of obligation' (Onuoha 2019). Less obvious is the fact that private practice reveals a strong correlation (0.71) with official religiosity compared to the moderate connection of this factor with folk religiosity. Some light might be shed on it in the perspective of the fact that private practice refers to the intensity of contact with transcendence and its subjective meaning (Zarzycka 2011). Official religiosity also has a greater correlation coefficient with the confession subscale of the MPI compared to the folk practices scale. It might be connected to the fact that the attitude represented by higher levels of OR refers to the positive perception of the sacrament of reconciliation. The latter presupposes the confession of sins. At the same time, public piety lays less emphasis on the sacramental aspect of the confession of sins and opens alternative ways for those who cannot receive the sacramental forgiveness of sins (Healy 2014).

The connection of OR to reception (from MPI) is stronger than that of FP. It shows that passive waiting for God's wisdom and guidance is closer to official and liturgical forms of Catholic practices. It is congruent with the fact that the official teaching of the Church emphasizes that the grace of God is something that a person receives through the offering of Christ rather than personal effort (Murray 2020). At the same time, folk piety is more of a bottom-up form of religiosity and through this, presupposes the activity on the part of the believer activates the meriting of God's attention, reducing the attitude of passive waiting (Francis 2014; Ryan 2012). In addition, the intellect subscale of CRS (that refers to the cognitive aspects of religiosity) reveals a stronger connection to OR than to FR. It shows that people preferring official, liturgical forms of Catholic practices are more eager to reflect

on the theological content of their faith and are more prone to pursue religious information (Huber and Huber 2012).

No significant differences were found in the correlation of both CRPQ subscales with thanksgiving and supplication. It may suggest that both these aspects of prayer—giving thanks and making requests to God—are equally present in the official and folk forms of Catholic religious practices.

As the scale developed in the presented study aimed to measure specifically Catholic religious practices, it adheres to the indications formulated by researchers to take into account the distinctive characteristics of a particular religious tradition. Hill (2013) puts it explicitly: "Researchers are encouraged to develop measures indigenous to the population or culture of interest" (p. 53). Although some items in the CRPQ, for example, fit well with other Eastern conceptions of religiosity (e.g., "The religiosity of the people is testimony to the faith of people with simple hearts"), other items reflect the religious particularity of Catholicism (e.g., "I try to attend Mass every Sunday"). By creating a scale that taps accurately into the complexity of Catholic religious practices, the authors provide an opportunity for psychologists to combine research with other social and humanities scientists (e.g., cross-cultural psychologists, sociologists, theologians) in order to reveal associations between religious behavior and a broad spectrum of psychological, social, and cultural factors. In this sense, our questionnaire follows in the line of previous tools that measured Christian religious practices, rituals, and behaviors from a Catholic perspective (Büssing et al. 2014, 2017). It is also in line with a recent research trend to develop scales that examine religious practices and behavior within a specific religion, e.g., Islam (Aziz et al. 2021) or Hinduism (Jayakumar and Verma 2021). In addition, data related to the role of popular piety and social change (such as the preservation of national identity during the partitions of Poland and World War II), the overthrow of communism, and current secularization trends provide a case for studying the role of folk practices among Polish Catholics (Roszak and Tykarski 2020).

## 6. Limitations

The limitations of the present study are related to the cross-sectional and correlational research model. Future research should consider the dynamics of changes in the intensity of Catholic religious practices and their impact on parameters related to psychological well-being and perceived physical health. The second limitation relates to the selection of the sample, which was not random. Although the normality of the distribution of the variables was verified, a representative group would have to be surveyed in order to be able to generalize the results. Another limitation refers to the potential fluctuations in people's religious practices due to unexpected events (e.g., deconversion, religiously oriented traumatic experiences). Longitudinal studies using the CRPQ are, therefore, advisable to examine the potential transformations of religious practices over a lifetime (taking into account developmental changes, such as cognitive or those related to the moral development). Subsequent limitation concerns the lack of variables showing a broader psychological context (referring, for example, to the personality-related determinants of the types of Catholic religious practices). This may allow for a better understanding of the constructs measured by the CRPQ. Another limitation is related to the use of reverse-coded items. It raises the question whether using antonymic expressions in the context of official religiosity allows us to measure accurately this psychological construct. The last limitation that the authors are conscious of is the fact that the purpose of this study was to create a brief and statistically accurate measure. For this reason, a difficulty arose in the selection of items that was reflected in the face validity of the tool. This reinforces the need for further research and verification in this area.

## 7. Conclusions

This article describes the construction and initial validation of a new 10-item tool for measuring Catholic religious practices (See Appendix A). According to the authors'

expectations, a two-factor structure has been successfully established through EFA, differentiating two dimensions: (1) official religiosity and (2) folk practices. Both subscales found confirmation through CFA on the basis of the data from the second study. Of essential importance is that both of the new subscales presented evidence for internal reliability and convergent validity. While remaining a specific measure of the Catholic form of faith manifestation, the Catholic Religious Practices Questionnaire correlates positively with commonly used general measures of religiosity, preserving the unique relationship between the two factors. Although the authors are convinced that CRPQ might be a useful tool for research and practice, there is a necessity for further tests on the reliability and validity of the new measure.

**Author Contributions:** Conceptualization, D.K., M.S., A.F. and J.L.; methodology, D.K., M.S. and A.F.; formal analysis, D.K., M.S., A.F. and J.L.; investigation, D.K., M.S., A.F. and J.L.; resources, D.K., A.F. and J.L.; data curation, D.K. and J.L.; writing–original draft preparation, D.K., M.S., A.F. and J.L.; writing–review and editing, D.K., M.S., A.F. and J.L.; supervision, D.K. and J.L. All authors have read and agreed to the published version of the manuscript.

**Funding:** The Article Processing Charge was funded by the last author's university.

**Institutional Review Board Statement:** The project was conducted according to the guidelines of the Declaration of Helsinki and approved by the Research Ethics Committee of The John Paul II Catholic University of Lublin (Institute of Theological Sciences), protocol code, KEBN_31/2022).

**Informed Consent Statement:** Informed consent was obtained from all participants involved in the study.

**Data Availability Statement:** The data that support the findings of this study are available from the corresponding author, M.S., upon reasonable request.

**Conflicts of Interest:** The authors declare no conflict of interest.

## Appendix A

### Kwestionariusz Katolickich Praktyk Religijnych (KKPR)

Poniżej znajdują się twierdzenia dotyczące Pani/Pana opinii na temat religijnych praktyk wiary katolickiej pod postacią liturgii oraz innych praktyk. Przez liturgię rozumiemy Mszę Świętą i sakramenty (np.: chrzest, bierzmowanie, namaszczenie chorych). Jako pozostałe praktyki katolickie rozumiemy formy pobożności stosowane poza liturgią (np.: odmawianie różańca, koronki do Bożego miłosierdzia, śpiewanie Litanii Loretańskiej, pielgrzymki do sanktuariów maryjnych itp.). Nie ma tutaj odpowiedzi dobrych ani złych. Zależy nam wyłącznie na poznaniu Pani/Pana opinii.

Proszę uważnie przeczytać każde z poniższych twierdzeń, zwracając uwagę na jego treść i określić, w jakim stopniu zgadza się Pani/Pan z nim, zakreślając wybraną cyfrę spośród znajdujących się obok twierdzenia. Poszczególne cyfry oznaczają:

1—zdecydowanie nie zgadzam się
2—nie zgadzam się
3—raczej nie zgadzam się
4—nie mam zdania
5—raczej zgadzam się
6—zgadzam się
7—zdecydowanie zgadzam się

| | | | | | | | |
|---|---|---|---|---|---|---|---|
| 1. Unikam chodzenia na Mszę świętą, podczas której księża straszą piekłem i karą za grzechy. * | 1 | 2 | 3 | 4 | 5 | 6 | 7 |
| 2. Formy pobożności ludowej w prosty sposób wyrażają wiarę w Boga. | 1 | 2 | 3 | 4 | 5 | 6 | 7 |
| 3. Pobożność ludowa jest prawdziwym wyrazem wiary chrześcijańskiej. | 1 | 2 | 3 | 4 | 5 | 6 | 7 |
| 4. Wszystkie religie proponują podobne wartości, co świecki światopogląd. * | 1 | 2 | 3 | 4 | 5 | 6 | 7 |
| 5. Udział we Mszy świętej jest dla mnie najważniejszym elementem praktykowania wiary. | 1 | 2 | 3 | 4 | 5 | 6 | 7 |
| 6. Ludowe praktyki religijne przyczyniają się do wzrostu w wierze. | 1 | 2 | 3 | 4 | 5 | 6 | 7 |
| 7. Zasady moralne proponowane przez Kościół nie przystają do dzisiejszej rzeczywistości. * | 1 | 2 | 3 | 4 | 5 | 6 | 7 |
| 8. Proste, popularne praktyki religijne wyrażają głębię wiary. | 1 | 2 | 3 | 4 | 5 | 6 | 7 |
| 9. W miejscach publicznych (np.: gminy, placówkach użyteczności publicznej, szkołach publicznych i szpitalach) nie powinno się umieszczać symboli religijnych, takich jak krzyże czy wizerunki Najświętszej Marii Panny. * | 1 | 2 | 3 | 4 | 5 | 6 | 7 |
| 10. Pobożność ludowa jest świadectwem wiary ludzi prostego serca. | 1 | 2 | 3 | 4 | 5 | 6 | 7 |

* Pozycje z odwróconą punktacją (1 = 7, 2 = 6, 3 =5, 4, 5 = 3, 6 = 2, 7 = 1)
Praktyki liturgiczne: 1*, 4*, 5, 7*, 9*
Pobożność ludowa: 2, 3, 6, 8, 10

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
