# Peer review of "Catholic Religious Practices Questionnaire (CRPQ): Construction and Analysis of Psychometric Properties"

_religions, doi:10.3390/rel13121203_

Round 1

Reviewer 1 Report

Thank you for the opportunity to review this manuscript.

The article concerns the development and validation of the "Catholic Religious Practices Questionnaire" tool, which was presented in the description of two studies. The author presents a tool with good psychometric parameters that can be used in scientific research, especially those concerning Catholic religious tradition.

In my view, the possibility of measuring Catholic religious practices could be beneficial. For example, it would be valuable to check how the frequency of religious practices relates to spirituality and life satisfaction or whether the frequency of religious practices translates into help in dealing with difficult situations.

Overall, I believe that the present work has its merits as it tackles an important question. At the same time, I have some comments regarding the reviewed article:

1. I suggest changing the structure of the article a bit. In the "1. Introduction" section, you could add a paragraph describing the purpose of the study. Then I would include a section "2. Theoretical background" which would consist of: "2.1. Specific Features of Catholic Religious Practices" and "2.2. Psychological Measurement of Religious Practices".

2. In the introduction, please define more clearly what official religiosity and folk practices are.

3. Please specify whether the group in studies 1 and 2 was Polish. If so:

4. The tool included in the appendix to the article is in the English version, while the development and validation of this tool was carried out with the participation of a Polish, not an English-speaking population. Therefore, the validated tool should be attached to the presented work in Polish, so that readers can use it in their research with the participation of Poles.

5. Please expand the description in the discussion on the importance of conducting research on the Polish group for the folk practices (FP) subscale.

6. OR items are mostly reverse scored, I would at least write a comment in the limitations as to what consequences this may have.

7. Perhaps the text would have benefited if, in study 2, you had explored alternative models, e.g., a one-factor model.

Author Response

Responses to Reviewer 1

The authors highly appreciate the Reviewer’s efforts to carefully review the paper and the precise comments which were very beneficial for revising our article. Thank you for both factual and formal suggestions. 

The following changes were introduced in accordance with the Reviewer’s comments:

  1. I suggest changing the structure of the article a bit. In the "1. Introduction" section, you could add a paragraph describing the purpose of the study. Then I would include a section "2. Theoretical background" which would consist of: "2.1. Specific Features of Catholic Religious Practices" and "2.2. Psychological Measurement of Religious Practices".

– In line with the suggestion, we changed the structure of our article. An additional paragraph describing the purpose of the study was added in the "1. Introduction" section. We stated that the aim of the present study is to develop a reliable and valid questionnaire that would assess Catholic religious practices conceptualized within the dogmatic and liturgical norms of the Catholic Church.  Within this objective, we will assess the psychometric qualities of the questionnaire: the factor structure through Exploratory Factor Analysis, the association between manifest variables and latent factors through Confirmatory Factor Analysis (CFA), internal reliability and convergent validity.

The section "2. Theoretical background" was also corrected as follows: "2.1. Specific Features of Catholic Religious Practices" and "2.2. Psychological Measurement of Religious Practices".

  1. In the introduction, please define more clearly what official religiosity and folk practices are.

– At the end of Introduction in 2.3. Overview of the Present Research we defined official religiosity and folk practises. We stated that official religiosity represents the extent to which believers shows compliance with the demands of the Catholic Church in terms of practice and observance of the official teaching and the acceptance level of religious symbols and teachings in the public space. Folk practices can be defined as the subjective importance of customary but informal forms of prayer called popular Catholic piety.

  1. Please specify whether the group in studies 1 and 2 was Polish. If so:

 - Yes, it is true, the sample consisted entirely out of Polish participants.

  1. The tool included in the appendix to the article is in the English version, while the development and validation of this tool was carried out with the participation of a Polish, not an English-speaking population. Therefore, the validated tool should be attached to the presented work in Polish, so that readers can use it in their research with the participation of Poles.

– Thank you for this suggestion. Our intention was to make the tool more comprehensive for English-speaking reader and encourage to make an English validation. But we understand that attaching an original version, which was validated, can be beneficial for further examination of psychometric properties of the tool, so we added it the Appendix.

  1. Please expand the description in the discussion on the importance of conducting research on the Polish group for the folk practices (FP) subscale.

– We have not specifically hypothesized on the uniqueness of Polish profile of folk practices, but in line with the suggestion we added the following paragraph to the discussion:

“In addition, data related to the role of popular piety and social change (such as the preservation of national identity during the partitions of Poland and World War II), the overthrow of communism, and current secularization trends provide a case for studying the role of folk practices among Polish Catholics (Roszak and Tykarski 2020)..”

  1. OR items are mostly reverse scored, I would at least write a comment in the limitations as to what consequences this may have.

– Although we have justified the use of reverse-coded items as a mean to avoid response style bias we agree with the Reviewer that it might cause certain dangers. Our reverse coded items were not created through the strategy of simple negation, but were formulated by using an antonymic expression. Still, it raises question if it allows to measure same psychological construct as items that are “positively” coded. Statistical analyses show a good fit even though referring to this method. But checking it on other populations might be decisive for evaluating such a solution. We have supplemented with a relevant explanation in the limitations section:

“Another limitation is related to the use of reverse-coded items. It raises the question whether using antonymic expressions in context of official religiosity allows to measure accurately this psychological construct.”

  1. Perhaps the text would have benefited if, in study 2, you had explored alternative models, e.g., a one-factor model.

– We tested a 3-factor model, considering that folk practices are a multifaceted concept. For the purpose of analysis, we called the 3rd factor Popular faith. However, the statistics showed that the 3-factor model represented the data less adequately than 2-factorial model. We added this information in our revision on p. 10.

Reviewer 2 Report

The paper presents an interesting topic - a new questionnaire concerning catholic religious practices. The paper is written with high quality and is very well methodologically conducted. The results of both EFA and CFA are well presented, and the results are even confirmed by correlations with another research instrument (MPI).

Only one thing should be discussed and explained more deeply: selecting items for the pool. The authors developed 40 items pool and asked four experts to reduce it. How the items were created? Were they based on known items from different measurement instruments, or were they originally created by the authors? Were there any criteria specifying the creation? In the case of expert selection, what were the criteria for selection? How and on which basis (if any?) were the 40 items compared to each other? How was the process of selection conducted? I recommend just adding a short explanation of the items selection process for the cleanliness of the whole methodological process.

Author Response

Responses to Reviewer 2

The authors highly appreciate the Reviewer’s efforts to carefully review the paper and the precise comments which were very beneficial for revising our article. Thank you for both factual and formal suggestions. 

The following changes were introduced in accordance with the Reviewer’s comments

1) Only one thing should be discussed and explained more deeply: selecting items for the pool. The authors developed 40 items pool and asked four experts to reduce it. How the items were created? Were they based on known items from different measurement instruments, or were they originally created by the authors? Were there any criteria specifying the creation? In the case of expert selection, what were the criteria for selection? How and on which basis (if any?) were the 40 items compared to each other? How was the process of selection conducted? I recommend just adding a short explanation of the items selection process for the cleanliness of the whole methodological process.

– We clarified the process, trying to include all the suggestions. We stated that Directory on Popular Piety and Liturgy was one of the main sources of inspiration for creating items. In addition, we included people’s views on their practice of faith. Due to the specificity of the planned questionnaire, we did not use other statements from the existing scales. In this sense, all items were originally created by the research team. The main criterion adopted in the creation of items was a reference to the manifestations of official religiosity (e.g., liturgy) and folk piety (e.g., cultural elements), understood as complementary forms of expressing faith.

We also added that four experts in the field of theology and psychology examined congruency of items. They evaluated all items considering the compliance of the content of each statement with the criterion referring to official religiosity and popular piety.

Reviewer 3 Report

I have reviewed the manuscript, titled “Catholic Religious Practices Questionnaire (CRPQ): Construction and Analysis of Psychometric Properties” (religions-2074390). The aim was to present the construction and initial validation for a brief questionnaire to measure Catholic religious practices.

The study is a very good piece of psychological research (clear introduction, accurate statistical analysis, constructive discussion).

However, I ask the authors to address some points:

1) The sentence on p. 2, lines 67-68: "From a sociological point of view, religious practices are regarded as the religious behaviours of members of a religious group (Day 2020)" needs to be made more specific. As the scale is intended for the Catholic faith, please list the most important religious practices in the Catholic Church
2) On p. 11, after lines 385-387, there should be added what is meant by high and low scores on the subscale official religiosity (OR). Similarly on p. 11, after lines 406-408, the same should be done in relation to the subscale folk practices (FP).
3) In Limitations, p. 13, what specific longitudinal studies could be proposed to examine potential transformations of religious practices over a lifetime?
4) In References, some publications have doi, whereas others do not – it needs to be standardized.
5) In References, there are two articles by Głaz. 2021. Please, add ‘a’ and ‘b’ to distinguish them.

Author Response

Responses to Reviewer 3

The authors greatly appreciate the Reviewer’s efforts to carefully review the paper and the valuable suggestions offered. They were very helpful and instrumental in revising our article and improving its quality.

The following changes were made in accordance with the Reviewer’s comments:

1) The sentence on p. 2, lines 67-68: "From a sociological point of view, religious practices are regarded as the religious behaviours of members of a religious group (Day 2020)" needs to be made more specific. As the scale is intended for the Catholic faith, please list the most important religious practices in the Catholic Church.

– In response to this suggestion, we added the examples of the most important religious practices in the Catholic Church: Holy Mass, Devotions to Jesus Christ (e.g. Eucharistic adoration, the Divine Mercy Devotions), novenas, various litanies (e.g. to Our Lady, to saints), the Stations of the Cross, prayers (e.g. the Angelus, the Rosary), Devotions to saints (e.g. prayers of intercession).

2) On p. 11, after lines 385-387, there should be added what is meant by high and low scores on the subscale official religiosity (OR). Similarly on p. 11, after lines 406-408, the same should be done in relation to the subscale folk practices (FP).

– We added explanation of high and low scores of the both subscales at the end of relevant paragraphs:

“High scores on the OR subscale mean intensive cognitive and behavioral appreciation of liturgy and official teaching of the Catholic Church. Low scores are connected to poorer attendance on Sunday Mass, lower observance of the teaching of the Church and affirmation of the laicization processes in public sphere.”

“High scores on the FP subscale show elevated appreciation of practice of the Catholic devotions and belief in the positive impact it has on the development of faith of the individual. Low scores mean rejection of spontaneous forms of piety and demeaning its role in the profile of prayer of a Catholic person.”

3) In Limitations, p. 13, what specific longitudinal studies could be proposed to examine potential transformations of religious practices over a lifetime?

– Our idea was to propose a longitudinal study related to the development of faith over the course of human development, taking into account cognitive and moral development from childhood, through adolescence, to adulthood (as well as through its stages). We added explanation to the text in the bracket “(taking into account developmental changes, such as cognitive or those related to the moral development)”.

4) In References, some publications have doi, whereas others do not – it needs to be standardized.

– In order to standardize the References we removed the doi when necessary.

5) In References, there are two articles by Głaz. 2021. Please, add ‘a’ and ‘b’ to distinguish them.

– We added the letters of ‘a’ and ‘b’ to the two articles of Głaz into our revision.

Głaz, Stanisław. 2021a. Psychological Analysis of Religiosity and Spirituality: Construction of the Scale of Abandonment by God (SAG). Journal of Religion and Health 60: 3545–3561.

Głaz, Stanisław. 2021b. Psychological Analysis of Religious Experience: the Construction of the Intensity of Religious Experience Scale (IRES). Journal of Religion and Health 60: 576–595.